# Combinatory Library of Microorganisms in the Selection of Reductive Activity Applied to a Ketone Mixture: Unexpected Highlighting of an Enantioselective Oxidative Activity

**DOI:** 10.3390/microorganisms11061415

**Published:** 2023-05-27

**Authors:** Sofiane Ali Rachedi, Maximillien Genest, Stéphane Mann, Didier Buisson

**Affiliations:** Unité Molécules de Communication et Adaptation des Microorganismes, Muséum National d’Histoire Naturelle, CNRS UMR 7245, CP54, 57 rue Cuvier (63 rue Buffon), 75005 Paris, France; sofianealirachedi@yahoo.com (S.A.R.); maxgenest@hotmail.fr (M.G.); stephane.mann@mnhn.fr (S.M.)

**Keywords:** whole cell screening, tandem biocatalysis, enantioselective oxidation, deracemization

## Abstract

Biocatalytic processes are increasingly used in organic synthesis for the preparation of targeted molecules or the generation of molecular diversity. The search for the biocatalyst is often the bottleneck in the development of the process. We described a combinatorial approach for the selection of active strains from a library of microorganisms. In order to show the potential of the method we applied it to a mixture of substrates. We were able to select yeast strains capable of producing enantiopure alcohol from corresponding ketones with very few tests and highlight tandem reaction sequences involving several microorganisms. We demonstrate an interest in the kinetic study and the importance of incubation conditions. This approach is a promising tool for generating new products.

## 1. Introduction

Enzymes are catalysts of choice for many reactions of organic chemistry and the biocatalytic toolbox can be enriched by highlighting new reactions via enzyme discovery or protein engineering [1]. Bioconversions are used in chemistry for two purposes, either for the preparation of a target product in a synthetic scheme or for the generation of molecular diversity from an active molecule. The latter is applied to respond to various objectives, such as preparing the mammalian metabolites of a drug or increasing the biological activity of a natural product. In all cases, the selection of enzymatic activities remains a crucial step in the implementation of a bioconversion.

Microorganisms are the main source of enzymes, which can be used either in the purified form or via whole cells. For the latter, screening of active strains involves cell culture followed by incubations with the substrate to be transformed and the monitoring of the reactions. The conventional approach is relatively laborious and time-consuming in that these different steps are performed separately for each microorganism. Various methods have been developed to optimize the screening process, acting at the culture stage of microorganisms, the incubation conditions and/or at the follow-up stage of biotransformation. For example, microbial cultivation in microplates with liquid medium [2,3] or solid medium [4] allowed the steps to be automated. Recently, a high-throughput screening system using a nanoliter-range droplet-based microfluidic module to detect excreted enzymes has been described [5]. Another approach is to perform a first sorting using selection pressure to obtain enriched cultures from environmental samples, for using the substrate to be transformed as the sole source of carbon [6,7] or nitrogen [8,9]. Concerning reaction monitoring, the implementation of methods to detect the product by colorimetry [10,11] allowed us to avoid time-consuming chromatographic analyses.

In order to develop an effective method of screening, we reported a method to test microorganisms in mixtures. Our previous study on the biotransformation of guttiferone showed that a combinatorial approach in the composition of mixtures could reduce the number of assays. With this strategy, the microorganisms are tested in a series of twelve, which are grouped into four mixtures, each containing five microorganisms. Under these conditions, the active yeast was selected by performing only four incubations instead of the twelve needed in the traditional approach [12]. This approach was used to test 24 microorganisms (16 fungi and 8 bacteria) on their ability to transform two sinomenine derivatives, and it allowed us to select active microorganisms by performing 8 incubations for one and 10 for the other derivative [13]. Since then, other studies have been successfully carried out with this approach: one concerns the fate of furosemide in the environment [14], where four active strains were selected among seventy-two in only eight assays, and the other is about the search for molecular diversity from ilimaquinone, a molecule with various biological activities [15].

We show here that this screening method based on a combinatorial library of microorganisms (CLM) can be improved by its implementation on mixtures of substrates. The results obtained in the selection of reductive activity toward a mixture of two ketones demonstrate the interest in this approach since the reduction rates are the same as those observed during experiments with ketones alone. We also show that screening by CLM allows us to highlight non-targeted modifications resulting from the sequential activity of different strains. For example, we have selected a strain capable of selectively oxidizing (S)-phenylethanol, which was the product of the reduction of acetophenone by another strain.

## 2. Materials and Methods

### 2.1. Analytical Methods

GC analyses were performed on an OV1701 (Pierce, 15 m × 0.20 ram) capillary column, with helium (10 Psi) as a carrier gas and with gradient temperature for mixtures 1 and 2: 90 °C (5 min), then 90–120 °C (5 °C/min) and 120 °C (3 min) (Appendix A); and for mixture 3: 90 °C (4 min), then 80–200 °C (8 °C/min) and 200 °C (10 min).

Enantiomeric excess was determined by GC analysis using column CP-chirasil-Dex CB (25 m 0.32 mm, 25 µm) and carrier gas He (15 Psi). For phenylethanol: oven temperature 100 °C. Retention times (*R*)-isomer 12.6 min, (*S*)-isomer 14.3 min (Appendix A). For phenylpropanol: 100 °C (10 min), then 100–120 °C (3 °C/min) and 120 °C (5 min). Retention times (*R*)-isomer 18.8 min, (*S*)-isomer 19.5 min.

### 2.2. Screening of Microorganisms

Cultures were maintained on agar slants (yeast extract 5 g/L, malt extract 5 g/L, glucose 20 g/L and agar 20 g/L) and stored at 4 °C. Liquid YMS medium containing (per L) glucose 16 g, yeast extract 4 g, malt extract 10 g and soybean peptones 5 g was sterilized without glucose at 121 °C for 20 min. Separately sterilized glucose solution was added afterward.

Yeasts were cultured for 48 h in 60 mL of culture medium at 30 °C in 100-mL erlenmeyer and put in an orbital shaker (200 rpm). Biomasses were collected by centrifugation (4000 rpm, 15 min), weighed (Appendix A), and then suspended in 12 mL potassium phosphate buffer (0.1 M, pH 7) for yeasts present in one batch and 24 mL for yeasts present in two batches. The combinatorial libraries of microorganisms were conducted using incubation batches (A–H) obtained by placing 4 mL of cell suspension as described in Table 1. Three mixtures of substrates were tested by the addition of solutions of ketones in isopropanol (0.2 mL) to obtain 1 g/L as the final concentration. In all cases, control experiments were performed by incubating each ketone individually with the same batches. Biotransformation was carried out in an orbital shaker at 30 °C and 1 mL of reaction was withdrawn after 4 h and 24 h, respectively, extracted with ethyl acetate (0.5 mL) and analyzed by GC. 

### 2.3. Biotransformation by Rhodotorula buffoni

*Rhodotorula buffoni* was cultured in YMS medium as described above, and after harvest, the biomass was resuspended in potassium phosphate buffer (0.1 M, pH 7). For incubations in anaerobic conditions, the buffer is previously saturated with nitrogen by a bubble for 10 min. Solutions of acetophenone and phenylethanol in isopropanol were added to the buffer (1 mL/100 mL buffer) to obtain desired final concentrations. 

## 3. Results

### 3.1. Screening of Active Strains

We tested the CLM method in the screening of active yeasts in the biotransformation of ketone mixtures in order to obtain corresponding alcohol. Two combinatorial libraries were set up and tested on three mixtures of two substrates, an aliphatic ketone and an aromatic ketone (Figure 1). The choice of ketones is based on the wish to test a mix of simple substrates, but sufficiently different for one of the substituents to minimize the risk of interaction/inhibition. Incubations were implemented according to a two-stage procedure. After growth, the biomass of each strain is collected by centrifugation and suspended in a buffer. These conditions make it possible to obtain fresh cells in quantity and avoid the risk of growth competition that would occur in a one-pot culture of several strains. The CLMs were formed randomly and the compositions of batches are described in Table 1. After the substrates were added, monitoring of the biotransformation and the comparison of reduction rates obtained from different batches allowed us to select the active strains. In order to show that the combinatorial selection method can be implemented on a mixture of substrates, it was therefore necessary to investigate whether the presence of one molecule affects the biotransformation of the other. We tested a mixture of ketones and each ketone was also incubated individually as controls. The results are given for two incubation time points used (4 and 24 h).

#### 3.1.1. Biotransformation with CLM-1

We studied CLM-1 in the biotransformation of mixtures of two methyl ketones (2-octanone **1** and acetonaphthone **3**) and of two ethyl ketones (3-octanone **2** and propiophenone **4**), mixture 1 and mixture 2, respectively. We observed (Figure 1) that the reduction rates in the assays are close to those observed in the controls, except for the reduction of 3-octanone in the presence of propiophenone with batch B. Indeed, after 24 h of incubation, we noticed a significant reduction (90%) of the control and a smaller reduction (51%) of the assay.

Comparison of yields obtained with the two aliphatic ketones show that the best reduction is observed in batch D, which makes it possible to select the strain *Yarrowia lipolytica* since this yeast is exclusively present in batch D. Concerning biotransformation of propiophenone and acetonaphthone, we observed a low reduction in batches B for assays and controls. The slow reaction rate may be due to the low reducing activity of the five strains in batch B, since these strains were excluded.

A large reduction with batch A and D in controls and assays was observed, and for both, an excellent enantiomeric excess of the corresponding alcohol was measured (>99%).

Consequently, two yeasts were good candidates, *Rhodosporidum turuloïdes* and *Pichia minuta*, since they are the only ones present in these two batches. The *R. turuloides* strain was shown to be effective since its incubation in the presence of propiophenone showed a 99% reduction rate after 24 h. The reduction of acetophenone by this yeast gives (*S*)-phenylethanol (76% yield, 91% ee).

A phenomenon involving the *Candida pinus* strain could be proposed to explain the low reduction of both aromatic ketones as it is present exclusively in batch B. Two hypotheses can be suggested: either the production by *C. pinus* of a molecule inhibiting the reduction of these ketones, or an effective oxidation of the corresponding alcohols by an enzyme produced by this strain. However, under resting cell conditions, cellular metabolism is slowed down with low metabolite production, thus making the first hypothesis of inhibition unlikely.

#### 3.1.2. Biotransformation with CLM-2

We studied CLM-2 in the biotransformation of aliphatic ethyl ketone (3-octanone **2**) in a mixture with aromatic methyl ketone (acetophenone **5**), mixture 3. The percentages of biotransformation (Figure 2) showed that there was little difference between the incubations of ketones alone (control) and those of mixture (assay) where no significant inhibition was observed.

The results obtained for the biotransformation of 3-octanone allowed us to select an active strain. We observed that the reduction reaction is very fast with batch E, with 85% and 82% of ketones reduced in the control and assay, respectively, after 4 h of incubation. It thus can be concluded that the yeast *Kluyveromyces dobzanskii* is responsible for this fast reduction.

In the case of the biotransformation of acetophenone, we observed unexpected results since the percentages of phenylethanol in batches E and F after 24 h incubation time were much lower (19–37%) than those observed after 4 h (70–79%), both in the control and the assay.

We deduced that the yeast involved in the transformation was *Cryptococcus macerans* and/or *Rhodotorula buffoni*. To understand the phenomenon, we studied the fate of phenylethanol in the presence of each of these two strains.

### 3.2. Biotransformation by R. buffoni

#### 3.2.1. Evidence of Oxidation of Phenylethanol by *R. buffoni*

Incubation of racemic phenylethanol (1 g/L) in the presence of yeast (100 g wet weight/L) was conducted for 24 h. Analysis of the incubation media showed no biotransformation with *C. macerans*, while with *R. buffoni* acetophenone (15%) was observed and the residual phenylethanol had *R*-configuration with an enantiomeric excess of 83% (Figure 2). These results demonstrated that *R. buffoni* was capable of oxidizing (*S*)-phenylethanol with excellent enantioselectivity.

However, these results were not consistent with only oxidation of (*S*)-phenylethanol since the yield of (*R*)-alcohol, from racemic mixture, is greater than 50%. So, in order to examine the redox reaction abilities of *R. buffoni*, we tested it on the reduction of acetophenone and the oxidation of phenylethanol in both anaerobic and aerobic conditions.

#### 3.2.2. Biotransformation in Anaerobic Conditions

At first, we wanted to know if the oxidation reaction depends on the aeration conditions of the incubation medium. The study of phenylethanol concentration during its incubation in anaerobic conditions showed that the alcohol was not modified and was not oxidized by yeast. Moreover, there was a reduction of acetophenone into (*S*)-phenylethanol, which was obtained with an enantiomeric excess of 57 to 66%, depending on the ketone concentration (Figure 3). These results showed that *R. buffoni* is also able to reduce acetophenone but in a mixture of enantiomers.

#### 3.2.3. Biotransformation in Aerobic Conditions

Biotransformation was conducted at 2 g/L (16 mM) substrate concentration and followed for 24 h. We observed (Figure 4) a rapid oxidation of phenylethanol since after 4 h, there remained less than 60% of alcohol, and then its proportion increased. After 24 h (*R*)-phenylethanol was obtained with a 69% yield and high enantiomeric excess (97%). On the contrary, the reduction of acetophenone was more limited than in anaerobic conditions since after 24 h phenylethanol was obtained with only 36% yield while it was 48% in anaerobic conditions. Moreover, the analysis of its stereochemistry and its enantiomeric excess was particularly interesting. Indeed, the main product obtained after 4 h of incubation time was the (*S*)-phenylethanol (7% ee) while it was the (*R*)-phenylethanol with 95% ee after 24 h. In these conditions, there is, concurrently, the reduction of acetophenone into a mixture of enantiomers and the oxidation of (*S*)-phenylethanol.

These results show that the reduction of acetophenone by *R. buffoni* leads to a mixture of enantiomers and can be catalyzed either by alcohol dehydrogenase (ADH) with little stereoselective or by two ADHs with opposite enantiotoposelectivities.

We then studied the effect of the initial concentration of substrate (Figure 5) on both reactions by measuring the amount of phenylethanol after 4 h of incubation, in order to limit the effect of the reverse reaction. It has been observed that, regardless of the concentration of alcohol, the oxidation reaction is rapid since there is already more than 40% conversion, which corresponded to the oxidation of a large part of (*S*)-phenylethanol. On the other hand, low phenylethanol formation was observed from acetophenone, which may be the result of its slow reduction or re-oxidation of alcohol.

The effect of biomass concentration was also investigated (Table 2). Oxidation of (*S*)-phenylethanol was observed to be less effective when the biomass concentration was 150 g/L compared to a concentration of 50 g/L, since after 24 h of incubation the proportion of alcohol is 88% and 69%, respectively. Reduction of acetophenone into (*S*)-phenylethanol was favored at high concentrations. Indeed, enantiomeric excess of (*R*)-phenylethanol not only was lower at 150 g/L (71%) than that obtained for a concentration of 50 g/L (95%) but the conversion percentage was also higher. These results can be explained by the decrease in the availability of oxygen for the oxidation of phenylethanol because its consumption is greater with the increase in cell concentration.

While the reductions are ADHs dependent, these results showed that oxidation of (*S*)-alcohol is oxygen dependent and suggests an oxidase activity (Figure 3).

## 4. Discussion

The selection of microorganisms capable of transforming a substrate is particularly interesting when the objective is not to obtain a targeted product but for the generation of molecular diversity for pharmacomodulation study or the study of xenobiotic metabolism. Indeed, in microbial biotransformation, especially in the process involving resting cells, cells are used as enzyme bags with various activities which can transform the substrate with different chemical reactions. This approach makes it possible to discover new enzymatic activities with the formation of unexpected products. For example, substitution on the quinone ring by ethanolamine, decyanation, oxidative intramolecular cyclization and Michael reaction was observed during the biotransformation of ilimaquinone [15], 1-benzylpyrrolidine-2,5-dicarbonitrile [16], guttiferone A [12] and hydroxychalcones [17], respectively.

A microorganism’s abilities in biotransformation processes are dependent on its growing and incubation conditions. It is therefore essential that these conditions are met in the implementation of active strain testing. The aim of this study was to show that the combinatorial selection method can be implemented on a mixture of reagents. This approach to testing substrate mixtures has been commonly used, especially when a large number of compounds are generated as in combinatory chemistry [18]. The incubation of a cocktail of substrates targeting different enzymatic activities has been already used to study the biocatalytic abilities of several microorganisms. This method allowed functional profiling to be obtained in one test for each strain [19]. To our knowledge, no biological or enzymatic activity has been sought by setting up a selection test on both a mixture of substrates and a mixture of microorganisms.

For the first study, we tested two structurally different ketones: aliphatic vs aromatic compounds. We observed no significant difference between assays and controls, which showed that there is no phenomenon of inhibition of one ketone on the other. This approach makes it possible to evaluate rapidly the global effectiveness of the strains for a substrate. For example, our results showed that 2-octanone is less well reduced than other ketones, and 3-octanone is very well reduced. It also appears that no selectivity of the strains applies regarding the aliphatic or aromatic nature of the substrate (Appendix A).

We have also shown that all the conditions are met for the search for as many reactions as possible. Thus, a series of reduction and oxidation reactions leading to the stereoinversion of secondary alcohol has been demonstrated in a few tests. Such biological stereoinversion allowed the preparation of enantiomerically pure molecules from meso compounds or racemic mixtures [20,21]. Two strategies have been developed, either with a single microorganism or with two microorganisms in tandem, one performing the oxidation reaction and the other the reduction reaction. Implementation of the first one is often the result of an unexpected stereoinversion observed during biotransformation, while the implementation of the second is the result of a selection of microorganisms having the required enzymatic activities.

Both were applied to the deracemization of (±)-phenylethanol and derivatives. Using the first strategy, (*R*)-phenylethanol was obtained by biotransformation involving the fungus *Geotrichum candidum* [22], while the production of (*R*)- and (*S*)-(*m*-fluorophenyl)ethanol was obtained from a racemic mixture by biotransformation performed with *Aspergillus terreus* strains [23]. Using the second strategy, (*S*)-phenylethanol has been obtained after isolating strains from the soil, the bacterium *Mycobacterium oxydans*, and the yeast *Rhodotorula* sp. which catalyzed oxidation and reduction, respectively [24]. Recently, enriched phenylethanol enantiomers were obtained using the yeast *Candida albicans* and the bacterium *Lactobacillus brevi*. These microorganisms were selected from a collection but no indication is given on the number of strains tested and the method used [25]. Using the conventional approach, one fungus has been selected through a screening of seventy-nine endophytic microorganisms on the reduction of acetophenone [26]. Two hundred and thirty-two environmental microorganisms were pre-selected using a culture media in which racemic phenylethanol was the sole source of carbon. They were evaluated individually on their ability to oxidize phenylethanol, and among them, fifty catalyzed the (*S*)-enantiomer oxidation [27].

Thanks to the CLM screening method, we were able, on one hand, to demonstrate reactions of ketone reduction into alcohol and re-oxidation of alcohol, and on the other hand, to select the yeast *R. buffoni* strain using six assays whereas the conventional method would have required twelve tests.

This was possible due to the conditions under which incubations were implemented. First, the assays were conducted on sufficient incubation volumes that resulted in sampling and monitoring of biotransformation. The following of the biotransformation of tetralones allowed Janeczko et al. to highlight the stereoinversion of resulting alcohols [28]. High throughput screening methods using micro-plates are not always suitable for monitoring biotransformations using multiple samples. Then, we showed that the oxidation reaction was oxygen dependent, it was able to take place during the test because the incubations were carried out in sufficiently oxygenated conditions, in a buffer (resting cells), with moderate cell concentration [29]. There is already mention of this dependence on the incubation medium oxygenation. Buisson et al. [30] reported the deracemization of ethyl (±)-3-hydroxybutanoate in 24 h by the fungus *Geotrichum candidum* if incubation was conducted in aerobic conditions. On the contrary, the transformation of (*R*,*S*)-hydrobenzoine into (*S*,*S*)-diol by the yeast *Saccharomyces uvarum*, one of the first examples of stereoinversion, was conducted in the growing medium with low levels of dissolved oxygen and was done in three weeks [31]. Similarly, the results obtained with the highest concentration of biomass may be due to the consumption of oxygen for the survival of yeast at the expense of the oxidation reaction. Finally, the yeast *R. buffoni* is able to reduce acetophenone but is not effective enough to explain the results obtained during screening with batches A and B. The significant reduction rate observed after 4 h of incubation time is due to one or other microorganisms present in lots A and B. This implies that the highlighting of the oxidative activity of *R. buffonii* was possible thanks to the CML approach used. So, we have identified a sequence of two reactions involving two microorganisms at least (Figure 6A). This finding is particularly interesting in an objective to generate a molecular diversity (Figure 6B). Indeed, we can consider obtaining new molecules through biotransformations involving several microorganisms. This approach may represent one aspect of the combinatorial biocatalysis, which is a powerful tool for the generation and optimization of lead compounds in drug discovery and development [32].

Finally, work is in progress to explain the results obtained during the incubation of propiophenone and acetonaphthone in batch B. Indeed, the evidence of oxidative activity in *C. pinus* as that observed in *R. buffoni* may be interesting, because of its effectiveness and enantioselectivity. It can be noted that these two strains have already been selected to catalyze the oxidative cyclization of phloroglucinol derivatives with the formation of xanthones.

## 5. Conclusions

We have demonstrated that the CLM screening method is effective in rapidly selecting active microorganisms and can be implemented on substrate mixtures. It significantly reduces the number of incubations and thus the number of chromatographic analyses. We tested 24 yeast on 5 substrates and 3 strains were selected for biocatalytic interest in 16 assays instead of the 120 needed via selection using a conventional approach. We were also able to highlight reactions not initially sought as the result of tandem reaction sequences involving several microorganisms. This could be an efficient method to generate molecular diversity since it can lead to the formation of original compounds not produced during one microorganism biotransformation (or during biotransformation with a single microorganism).

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
