# Peer review of "Combinatory Library of Microorganisms in the Selection of Reductive Activity Applied to a Ketone Mixture: Unexpected Highlighting of an Enantioselective Oxidative Activity"

_microorganisms, 2023, doi:10.3390/microorganisms11061415_

Round 1

Reviewer 1 Report

The authors develop an approach to easily identify the microorganism performing a reaction of interest. They do so by pooling different microorganisms in different batches with 2 strains overlapping in batch pairs and one strain specific to each batch. The authors have chosen to test each batch with a mix of an aliphatic and an aromatic substrate. They finally pursued the enantiospecific reaction of Rhodotorula buffonii on aromatic substrates.

It would be of interest for the authors to specify if ketone reduction to alcohol is important to industry or is just a proof of concept. The presentation is somewhat confusing at times. Authors do not explain the rational of there tests, certain batches are tested with some substrate mixes and others with others. Figures in text do not always point to the right one.

figure 1 and 2 are very hard to follow since in the material in methods they say controls are incubations with ketone alone. I think they meant with individual ketones. I think the results would be more legible if they substracted from each experiment with a mix the control value at the time point. In the case of 2-octanone, we would clearly see if acetonaphtone inhibits or not aliphatic ketone  reduction and would make stand any unexpected results.

I think the R. buffonii part well thought except that they refer in the text twice to figure 5 and not once to figure 6.

The authors do not discuss whether some microorganisms can reduce any aliphatic ketone or any aromatic ketone. Are their limits to the size of the chains? They do not discuss the reasons for forming the batches, is it random or is it based on prior knowledge. If it is the latter, the authors should mention the rational, if it is the former test batches from one set with the mix of the other set. It would also address my other question about the influence of the side chains on the process. Should it be largely independent of side chain length, it would increase the interest in the approach.

There are many google type translations in the text such as using mixture instead of mix or line 46 by coloring instead of colorimetry or line 4 highlighting instead of discovery. The text as a whole would benefit from the help of an English speaking since writer.

Reviewer 2 Report

This paper described the method of combining microbial libraries to screen highly active microbial communities that can catalyze the target reaction. This research method is novel and effective, and the author obtained ideal results by screening the combined library of microorganisms. This study has reference value for related research work.

1. The study is interesting, and the author cultured each microorganism separately before screening, which is very reasonable, and measured the biomass of yeast before mixed. But for the accuracy of screening and the repeatability of experimental results, it is recommended to supplement the biomass of each weight of microorganisms in the combined library before screening or the volume and OD values of each strain before mixing.

2. The rate of different strains in the combining microbial libraries is also an important parameter need to be considered. It is suggested to discuss it in the results and discussion section.

3. Other small points,

Line 60, please correct this sentence.

There are two 3.1.1.

The format the references is not correct, please check and correct them one by one.
